# Unusual prism adaptation reveals how grasping is controlled

Willemijn D Schot[†], Eli Brenner, Jeroen BJ Smeets*

Department of Human Movement Sciences, Vrije Universiteit Amsterdam, Amsterdam, Netherlands

**Abstract** There are three main theories on how human grasping movements are controlled. Two of them state that grip aperture and the movement of the hand are controlled. They differ in whether the wrist or the thumb of the hand is controlled. We have proposed a third theory, which states that grasping is a combination of two goal-directed single-digit movements, each directed at a specific position on the object. In this study, we test predictions based on each of the theories by examining the transfer of prism adaptation during single-digit pointing movements to grasping movements. We show that adaptation acquired during single-digit movements transfers to the hand opening when subsequently grasping objects, leaving the movement of the hand unaffected. Our results provide strong evidence for our theory that grasping with the thumb and index finger is based on a combination of two goal-directed single-digit movements.

DOI: https://doi.org/10.7554/eLife.21440.001

## Introduction

Two-digit grasping is a prototype of human motor coordination. Yet, there is an ongoing debate about how such movements are controlled (see for example *Kleinholdermann et al., 2007*; *Smeets et al., 2010*; *van de Kamp and Zaal, 2007*; *Zaal and Bongers, 2014*). Three main theories have been put forward. *Jeannerod (1984)* proposed that grasping consists of a transport component (movement of the wrist) that brings the hand to a location near the object and a grip component that ensures that the distance between the digits (grip aperture) changes in a manner that is appropriate for the size of the object. Others argue that the thumb is transported to a position on the object, with the index finger moving relative to the thumb to match grip aperture to the object's size (*Mon-Williams and McIntosh, 2000*; *Wing and Fraser, 1983*). Finally, we have proposed that the trajectories of the tips of the digits when grasping is simply the combination of the trajectories of the same tips in goal-directed single-digit movements ('pointing'), whereby each tip is moved to a specific position on the object (*Schot et al., 2011*; *Smeets and Brenner, 1999*; *Smeets and Brenner, 2001*; *Verheij et al., 2012*; *Voudouris et al., 2013*).

We have previously shown that people can simultaneously adapt movements of the finger and thumb of one hand to opposite prismatic displacements (*Schot et al., 2014*). In the current study, we let people grasp a cube and a cuboid before and after a period in which their single-digit movements were adapted in opposite directions. The three theories on the control of grasping make different predictions for how aftereffects of opposite adaptation during the single-digit movements will transfer to the *grip aperture* and the *grip position* during subsequent grasping movements (*Table 1*).

If, as *Jeannerod (1984)* suggested, grasping is controlled by planning the movement of the wrist and the opening and closing of the grip aperture, one would not expect opposite adaptation acquired during single-digit movements to transfer to grasping: the aftereffects of the opposite displacements of the wrist when adapting single-digit movements of the index finger and thumb will cancel each other during subsequent movements of both together when grasping, leaving the *grip*

*For correspondence:
J.B.J.Smeets@vu.nl

Present address: [†]Educational Development and Training, Utrecht University, Utrecht, Netherlands

Competing interests: The authors declare that no competing interests exist.

*position* unaltered. The *grip aperture* will also not be influenced by adapting pointing movements because it is irrelevant for single-digit movements.

If it is not the wrist movement that is controlled, but the movement of the thumb, as proposed by *Mon-Williams and McIntosh (2000)* and *Wing and Fraser (1983)*, adaptation of single-digit thumb movements should lead to an aftereffect in the transport of the thumb. This aftereffect will be in the opposite direction than the visual displacement of the thumb in the adaptation phase. The movement of the index finger relative to the thumb will remain unaffected, again because grip aperture is irrelevant during a single-digit movement. The prediction is thus that *grip position* will show the aftereffect adaptation of the thumb, without any alteration of *grip aperture.*

Our own theory, based on the control of single-digit movements, predicts transfer of adaptation during pointing movements to the *grip aperture,* but not to the *grip position,* because when single-digit movements that are adapted in opposite directions are executed simultaneously, the aftereffects will cancel each other for the *grip position,* but will summate for *grip aperture.* So, if movements of the index finger are performed under leftward visual displacement and movements of the thumb are performed under rightward visual displacement during adaptation (*thumb right*), the predicted aftereffect is a rightward deviation of the index finger and a leftward deviation of the thumb, leading to a larger *grip aperture.* When the visual displacements are reversed, *grip aperture* should decrease.

## Results

The adaptation of pointing with the tip of each of the two single digits (central part of *Figure 1A*) was similar to the equivalent adaptation in our previous experiment (*Schot et al., 2014*). After removing the prisms, the digits' movements during grasping showed a clear aftereffect of the adaptation during pointing. Interpreted in terms of a transport- and grip-component, we found a clear aftereffect of adapting to prisms during single-digit pointing movements on the *grip aperture* on subsequent grasping trials. In the *thumb right* adaptation, the target for pointing with the tip of the thumb was displaced to the right, and that for pointing with the tip of the index finger was displaced to the left. Adapting to these displacements made the thumb move more leftward and the index finger more rightward to a certain visual location. When done simultaneously, this should move the digits further apart. In accordance with our theory that grasping can be regarded as simultaneous movements of the individual digits, *grip aperture* did increase after *thumb right* adaptation (and decrease after *thumb left* adaptation), both for the small and the large block (*Figure 1B*). As the two digits were always adapted in opposite directions, we expected their average (which we use as a measure of *grip position)* to remain unaffected by the adaptation. Our results show that this is the case (*Figure 1C*).

Statistical analysis of the effects of session, adaptation phase, and target block size on *grip aperture* showed a significant phase by session interaction ($F(1,7)=7.3$, $p=0.03$) indicating that *grip aperture* indeed changed differently after *thumb left* and *thumb right* adaptation (*Figure 1B*). This difference is 1.2 cm. Besides the phase by session interaction, there were also significant effects of block size (the target cuboid was grasped with a 2.3 cm larger grip aperture than the target cube; $F(1,7)=324.1$, $p<0.001$) and of session ($F(1,7)=24.8$, $p=0.002$). Statistical analysis of the effects of session, adaptation phase, and target block size on the *grip position* did not show any significant effects (all $p>0.1$, *Figure 1C*). The difference corresponding to the non-significant tendency towards an

**Table 1.** Whether or not adapting the two digits to opposite prisms during pointing movements would be expected to influence *grip position* and *grip aperture* during subsequent grasping movements according to the three main theories about the control of grasping.

| | Expected influence on | |
| --- | --- | --- |
| Controlled in grasping | Grip position | Grip aperture |
| Grip aperture and wrist position | No | No |
| Grip aperture and thumb position | Yes | No |
| Finger position and thumb position | No | Yes |

DOI: https://doi.org/10.7554/eLife.21440.002

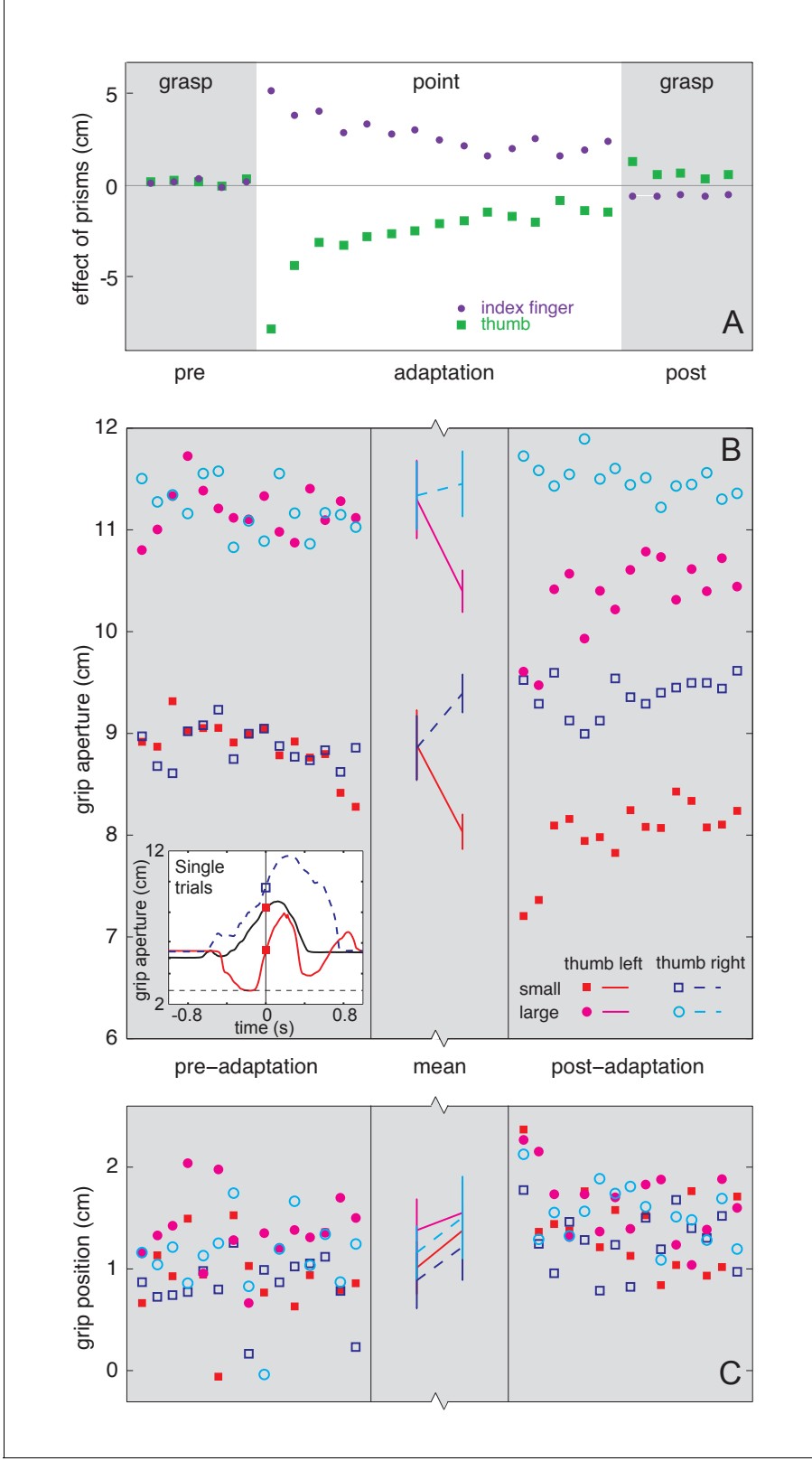

**Figure 1.** The prisms' effects on the individual digits, grip aperture and grip position. Overview of the individual digits throughout the experiment (**A**) and a detailed view of the grip aperture (**B**) and grip position (**C**) for the grasping movements performed in the pre- and post- adaptation phases. In panel A the symbols show the overall mean effect of the prisms on the position of the digits for each block of 6 trials during the three phases of the experiment. In the adaptation phase (white background), the digits partially adapt to the prisms during pointing movements. When switching to

*Figure 1 continued on next page*

*Figure 1 continued*

grasping movements under normal vision (grey background on the right) we see an aftereffect. In panels B and C, the symbols show the trial-by-trial time course averaged over all participants. The bars in the center of each panel show the average values (Mean ±SEM of the individual participants' median values within each phase; source file available as *Figure 1—source data 1*). A positive grip position is to the right. The inset shows one participant's time course of grip aperture during the last movement towards the target cube in one of the pre-adaptation phases (black curve) and during the first movements towards the target cube in both post-adaptation phases (dashed blue and solid red curves). Time 0 is the moment at which we determined the digits' positions for calculating grip aperture and grip position. The horizontal dashed line is the grip aperture at which the digits touch each other.

DOI: https://doi.org/10.7554/eLife.21440.003

The following source data is available for figure 1:

**Source data 1.** Positions at 1 cm before crossing the board for of all trials underlying Figure 1 and the summary data underlying the central parts of *Figure 1B,C*.

DOI: https://doi.org/10.7554/eLife.21440.004

adaptation phase by session interaction for *grip position* was only 0.1 cm, an order of magnitude smaller than the 1.2 cm that we found for *grip aperture*.

To give an indication of the relationship between our measure of *grip aperture* and the time course of changes in grip aperture *during* the movement, we plotted the time course of grip aperture during the last movement towards the target cube in one of the pre-adaptation phases and during the first movements towards the target cube in both post-adaptation phases for one representative participant (inset in *Figure 1B*). During the first movement towards the small block after *thumb left* adaptation (solid red curve), this participant's digits came so close together that they touched each other, giving her haptic feedback that they should be further apart. By the time her digits crossed the far end of the board, she had already partly corrected her movement. Thus, our measure of *grip aperture* gives an underestimation of the initial aftereffect. For *thumb right* adaptation something similar may have happened in some trials when the digits move so far apart that grip aperture reaches its anatomical limit, but this is not the case for the example in the dashed blue curve in the figure.

We designed our experiment around a set of predictions that could distinguish between three theories of grasping (*Table 1*), ignoring the joints and muscles involved in making the movements. One might argue that the ability to adapt in opposite directions with two digits is related to transfer more easily occurring from proximal to distal muscles than from distal to proximal muscles (*Krakauer et al., 2006*). We have ascertained that adaptation does not only transfer to grip formation, which primarily involves distal muscles, but also transfers to grip position, which primarily involves proximal muscles. We did this by adapting both digits in the same direction (see Appendix 1 'Transfer to grip position'). Although such transfer is not distinctive for our theory, it shows that adapting single-digit pointing movements does not only influence grip aperture, but involves the complete reach-to-grasp movement.

Some readers of our original adaptation study (*Schot et al., 2014*) have argued that the opposite adaptation of the two digits might be related to the eye rather than to the digit that was adapted, as we used a fixed digit-eye pairing during adaptation. If this objection would be substantiated, it might also affect the interpretation of the present study. We therefore decided to repeat the original experiment (with new participants in a new set-up, full description in Appendix 2 'Effect of digit-eye pairing') but using <u>monocular</u> single-digit pointing to measure the aftereffect. We did so using both the digit-eye pairing that we used in the adaptation phase and the opposite pairing. We replicated the results that we originally obtained with binocular viewing for monocular viewing. Most importantly, the aftereffect for each digit was equally large for both eyes, irrespective of whether the eye that was viewing was the one used to adapt the digit.

## Discussion

We show that adapting the thumb and index finger in opposite directions during single-digit movements gives rise to an aftereffect in *grip aperture* but not in *grip position* during subsequent grasping movements. Thus, the pattern of results is only consistent with the predictions based on our theory that grasping can be regarded as movements of the individual digits: adapting each digit

during single-digit pointing movements influences the grip aperture during grasping movements. The two other theories predict quite different patterns of results (*Table 1*). Transfer of prism adaptation has been shown to be absent or only partial when tested with a different movement speed (*Kitazawa et al., 1997*), throwing style (*Martin et al., 1996*) or load (*Fernández-Ruiz et al., 2000*). This shows that adaptation can be very specific to a particular movement (*van der Kooij et al., 2016*). The fact that we find transfer from single-digit pointing to the grip aperture in grasping is therefore not at all trivial but suggests that grasping is indeed the combination of two single-digit movements.

All three theories limit themselves to describing the movements of the task-relevant elements. They differ in what they consider to be the relevant elements for grasping. None of them is concerned with how the movements are achieved in terms of the roles of all muscles and joints involved, so none predicts how the many task-irrelevant elements are orchestrated (*Latash et al., 2002*; *van Beers et al., 2013*). Only the transport-grip theory is concerned with the movement of the wrist (used as a proxy for the position of the hand), which is therefore task-relevant according to that theory. We have previously noted that using the wrist as a proxy for grip position makes it necessary to add a third controlled variable because the position to which the wrist should be transported to grasp an object depends on the orientation of the hand and the size of the object (*Smeets and Brenner, 1999*). We therefore use 'grip position' (the average of the positions of the two digits) as a proxy for hand position.

How the wrist, elbow and shoulder move (or, more generally, how the redundancy problem is solved) is beyond the scope of any of the mentioned theories. We therefore interpret our findings in terms of the movements of the endpoints of the digits, completely ignoring how these movements are achieved in terms of muscles and joints. In doing so, we ignore the evidence that adaptation is specific for the joints involved in the pointing movements (*Krakauer et al., 2006*). Such evidence makes our findings even more surprising, because as can be seen in the example in *Video 1*, the participants mainly performed the pointing movements and their corrections from the elbow and shoulder, whereas changes in grip aperture must involve changing the angles of the digits relative to the hand. This suggests that the transfer that we found to grip aperture in grasping is not based on transfer of a joint-specific adaptation, but on a higher-level control of the trajectory of the digit in space.

Finding transfer from transporting the separate digits in pointing to grip aperture in grasping (*Figure 1B*) contradicts the idea that grip aperture is controlled separately from transport of the hand or thumb (see *Table 1*). Finding that grip position was not systematically further to the left after *thumb right* adaptation (*Figure 1C*) contradicts the theory that the thumb determines the transport of the hand. Both the presence of an effect on grip aperture and the lack of effect on grip position are consistent with similar but independent adaptation of the movements of finger and thumb. Finding that the grip aperture can increase as well as decrease depending on the direction of the adaptation shows that the effect is due to the adaptation itself, rather than for instance uncertainty about the size of the object leading to a larger hand opening.

The observation that the digits sometimes touch each other during the grasping movement (as in the example in the inset of *Figure 1B*) is also consistent with independent adaptation of the movements of finger and thumb. With independent control of the thumb and index finger, it is possible that people plan digit trajectories that cannot be executed simultaneously for anatomical reasons. This seems indeed to have happened in this study. Neither of the other theories can account for the digits touching each other during the grasping movement because if the aperture is controlled it is only expected to decrease (after reaching maximal grip aperture) until it matches the size of the object.

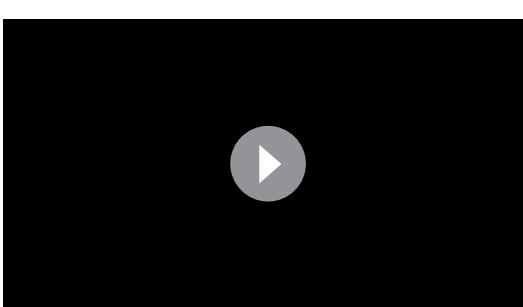

**Video 1.** Example trials. Three single digit trials (adaptation phase) and two grasping trials of one subject (pre- and post-adaptation phases). Note the presence/absence of the prisms, and the opening of the shutter glasses in front of one/both eyes.
DOI: https://doi.org/10.7554/eLife.21440.005

Together, these results provide strong evidence that grasping can be considered to be built up of the same movements as single-digit pointing: grasping is the combination of two goal-directed single-digit movements (*Smeets and Brenner, 1999*).

## Materials and methods

The methods employed in this study, are very similar to those of our original adaptation study (*Schot et al., 2014*). The main difference is the use of grasping instead of single-digit pointing in the pre- and post-adaptation phases. This study was part of a research programme that has been approved by the Ethische Commissie Bewegingswetenschappen (ECB 2006–02). All participants gave their informed consent before participating in this study.

### Procedure

Eight participants (25–29 years of age, two males) took part in the experiment after giving informed consent. They wore PLATO shutter glasses. Movements of Infrared emitting diodes attached to the fingernails of the thumb and index finger were recorded at 250 Hz using an Optotrak 3020 system. A target object was attached to a wooden board at one of three possible target locations (5 cm apart). The board obstructed the participants' vision of the hand until just before contact with the target object.

The experiments consisted of three phases: pre- and post-adaptation phases which each consisted of 30 grasp trials, and an adaptation phase which consisted of 30 pointing trials with the thumb and 30 pointing trials with the index finger. Three objects were used: a starting cube (2.3 cm), a target cube (2.3 cm), and a target cuboid (2.3 × 2.3×4.6 cm). The starting cube and the target cube were used in all three phases. The target cuboid was only used in the pre- and post-adaptation phases. It was included to force participants to use visual information to guide their grip aperture and to ensure that any observed adaptation effects are not specific to the object used in the adaptation phase.

Participants started all trials by grasping the starting cube with the glasses shut (*Figure 2, Video 1*). In the pre- and post- adaptation phases (i.e., on grasp trials), participants viewed the

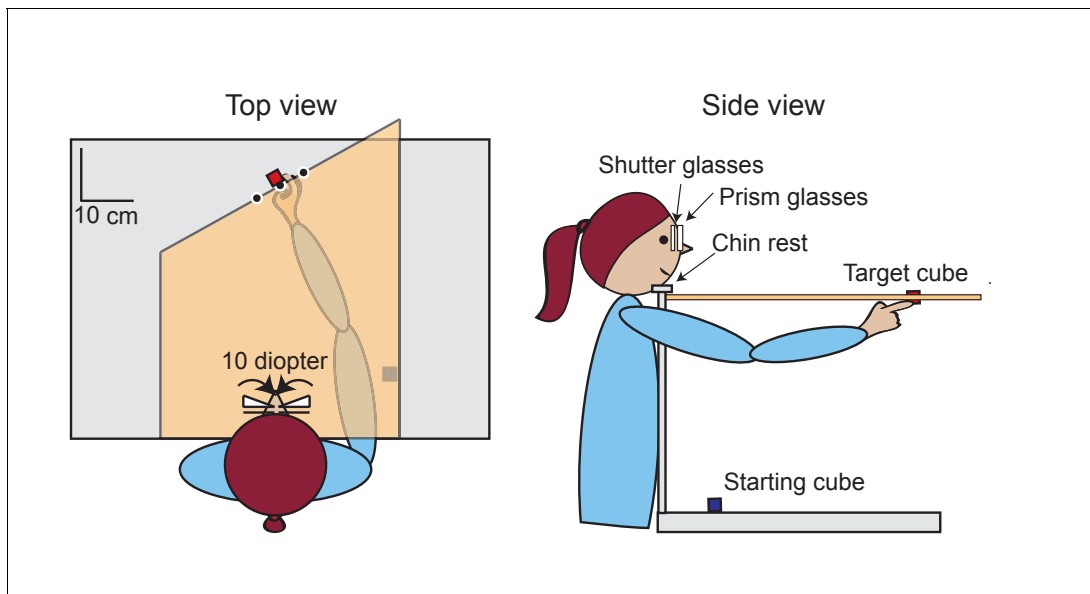

**Figure 2.** Top and side views of the experimental setup. The subject performs a single-digit pointing trial with the index finger. During single-digit trials, participants viewed the target block monocularly through a prism that was in front of the shutter glasses and moved their hand from the starting cube to touch the left side of the target cube with their thumb or to the right side of the target cube with their index finger. During grasping trials, participants were not wearing the prisms and viewed the target block binocularly. On these trials, participants grasped either the small target cube or the large target block (cuboid).

DOI: https://doi.org/10.7554/eLife.21440.006

target binocularly without prism glasses. Once the shutter glasses opened, they grasped either the target cube or the target cuboid by its left and right surfaces. The cuboid was oriented such that it was grasped by the surfaces that were twice as far apart as those of the cube. In the adaptation phase (i.e., on single-digit pointing trials), participants viewed the target monocularly through 10 diopter prism glasses that were worn over the shutter glasses. The prism in front of the left eye shifted the image of the target about 5 cm to the right. The prism in front of the right eye shifted the image of the target about 5 cm to the left. Before each pointing trial, participants were told either to touch the left side of the target cube with their thumb or to touch the right side of the cube with their index finger. Once the shutter glasses in front of the corresponding eye opened, participants moved the appropriate digit to the appropriate side of the target cube.

The experiment consisted of two sessions that were performed on separate days with their order counterbalanced across participants. In one session, vision was displaced to the right when touching the cube with the index finger and to the left when touching the cube with the thumb (*thumb left*). In the other session, the displacements were reversed (*thumb right*). Within each session, trials were presented in pseudo-random order that ensured that all trial-types were distributed evenly: in the pre- and post-adaptation phase, each combination of target size and target location was presented once every six trials; in the adaptation phase, each combination of digit and target location was presented once every six trials.

## Data analysis

To make sure that we had a measure that is not influenced by movement corrections based on visual feedback, we based our analysis for both index finger and thumb on the marker position 1 cm before it crossed the far edge of the board. We calculated the *grip aperture* by taking the difference between these positions for the two digits and the *grip position* by taking the average of these positions. As we used three target positions, *grip position* was expressed relative to the target's position.

We evaluated the adaptation effect by comparing the median value in the pre-adaptation phase with the median value in the post-adaptation phase, both for *grip aperture* and *grip position*. To test the predictions that follow from the different theories on grasping (*Table 1*), we performed two repeated measures ANOVAs with phase (pre-adaptation, post-adaptation), session (thumb left, thumb right), and target block size (small, large) as independent variables. One ANOVA had *grip position* as the dependent variable and the other had *grip aperture* as the dependent variable. A significant phase by session interaction would indicate that the adaptation during the pointing trials transfers to the grasp trials.

## Acknowledgements

This research was funded by NWO/MaGW [grant number 453-08-004]

## Additional information

### Funding

| Funder | Grant reference number | Author |
| --- | --- | --- |
| Nederlandse Organisatie voor Wetenschappelijk Onderzoek | 453-08-004 | Jeroen BJ Smeets |

The funders had no role in study design, data collection and interpretation, or the decision to submit the work for publication.

### Author contributions

Willemijn D Schot, Conceptualization, Investigation, Writing—original draft; Eli Brenner, Conceptualization, Writing—review and editing; Jeroen BJ Smeets, Conceptualization, Funding acquisition, Writing—review and editing

## Author ORCIDs
Eli Brenner ⓘ https://orcid.org/0000-0002-3611-2843
Jeroen BJ Smeets ⓘ http://orcid.org/0000-0002-3794-0579

## Ethics

Human subjects: This study was part of a research programme that has been approved by the Ethische Commissie Bewegingswetenschappen (ECB 2006-02). All participants gave their informed consent before participating in this study.

## Decision letter and Author response
Decision letter https://doi.org/10.7554/eLife.21440.012
Author response https://doi.org/10.7554/eLife.21440.013

---

## Additional files

### Supplementary files
• Transparent reporting form
DOI: https://doi.org/10.7554/eLife.21440.007

---

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

# Appendix 1

DOI: https://doi.org/10.7554/eLife.21440.008

## Transfer to grip position

Our theory predicts that adapting pointing movements of both digits in the same direction will transfer to the grip position in grasping. To test this, we performed a simplified version of the main experiment.

## Methods

The simplification consisted of using binocular vision throughout the experiment with leftward deviating prisms in front of both eyes during the adaptation phase, and using only a single target object. We furthermore reduced the number of trials in all phases of the experiment and limited the number of participants to six. All participants performed the experiment with their right hand, which was the dominant hand for five of them.

## Results

We found the expected rightward after-effect in the grip position of grasping when removing the leftward deviating prisms (*Appendix 1—figure 1*).

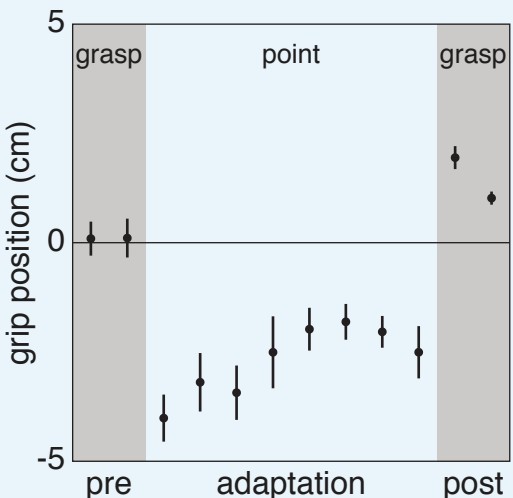

**Appendix 1—figure 1.** Data of the control experiment on the transfer of adaptation from single-digit pointing movements to the grip position in grasping. Participants started with 6 grasping trials (*pre*-adaptation; grey background). Subsequently, they made 24 pointing movements with either the index finger or thumb while wearing leftward deviating prisms (*adaptation*; white background)., Finally, participants performed 6 more grasping trials after removing the prisms (*post*-adaptation; grey background). Each dot is based on 3 trials per participant, and shows the average (±SEM) across participants. For the grasping trials, we first averaged the grip positions in the three trials for each participant. For the pointing trials, we first averaged the positions for each digit (if there was more than one value), and then averaged across index finger and thumb. If there were no values for one of the digits in the three trials, that participant's data does not contribute to that point.

DOI: https://doi.org/10.7554/eLife.21440.009

## Appendix 2

DOI: https://doi.org/10.7554/eLife.21440.010

## Effect of digit-eye pairing

Some readers of our original adaptation study (*Schot et al., 2014*) have argued that because we used a fixed digit-eye pairing during adaptation, the opposite adaptation of the two digits might be related to the eye rather than to the digit that was adapted. Although we regard it as unlikely that such an association could have influenced our results (as we used binocular vision for determining the aftereffect), we thought it would be wise to provide explicit experimental evidence against this explanation, because our interpretation of the transfer to grip aperture relies on the specificity for the digit rather than the viewing eye.

Our claim of simultaneous opposite visuomotor adaptation of the digits of a hand is independent of the type of perturbation and the exact nature of the goal-directed movements. We therefore took the opportunity to modify these aspects to ensure that the effects that we found are not specific to the task that we used as well as to make the protocol less labor-intensive than the original study (*Schot et al., 2014*). Instead of using prisms that displaced the view of physical targets that are touched at their sides, we used virtual targets in a horizontal plane and a cursor that provides erroneous terminal feedback that we expected people to adapt to. In order to evaluate any role of digit-eye pairing, we used monocular vision throughout the experiment. By adapting with a fixed pairing of digit and eye (with associated sign of the shift of the feedback), but testing with all four combinations of digits and eyes, we can determine whether the adaptation is indeed fixed to the digit (generalizing across eyes), and not specific to the adapted pairing of digit and eye (in which case we should observe no aftereffect for the digit-eye pairings that were not used during adaptation).

## Methods

Eight colleagues who were naïve with respect to the purpose of the experiment participated in this experiment. The set-up that we used was similar to that used in an earlier experiment (*Kuling et al., 2016*). The visual targets were black 2 cm diameter disks on a grey background. They were 20 cm from the starting position in the sagittal direction, and either aligned laterally with the starting position or 5 cm to the left or to the right. The target and background were projected onto a horizontal screen that was viewed by looking into a horizontal mirror below it. A board was positioned below the mirror, at the level of the image of the projection screen, so that the targets appeared to be lying on this board. Participants moved the index finger or thumb of their unseen hand to the visual targets. Movements were recorded by an Optotrak 3020. To initiate a trial, the participant positioned the specified digit on a physical, 4 cm diameter ring (the starting position) that was easy to find by touch alone on the otherwise flat, untextured surface. Participants wore Plato shutter spectacles that always occluded one of the eyes.

The experiment consisted of 2 sessions, one for each pairing of digit and eye. Each session consisted of 60 pre-adaptation trials (5 for each of the 3 positions, 2 eyes and 2 digits), 90 adaptation trials (15 for each of the 3 positions, and 2 combinations of eye and digit) and 60 post-adaptation trials that were identical to the pre-adaptation trials. Each trial started with a written instruction that specified which digit (index finger or thumb) had to move to the starting position. One of the Plato spectacles' shutters was clear and the other opaque. In the adaptation block, this was consistently coupled to the digit that was to be moved. In the other two blocks, it was random. The target appeared 500 ms after the relevant digit arrived at the starting position. Participants had to move to the target in a single smooth movement. Once the digit stopped moving (less than 1 mm displacement within 300 ms) its position was noted. 500 ms later the target disappeared. In adaptation trials, a 1.4 cm diameter red disk provided feedback about the digit's position during this last 500 ms. The feedback was 3 cm to the left or 3 cm to the right of the digit's actual position. Whether it was to the left or to the right was

coupled to the digit that was moving (and thus also to the viewing eye). The pairing differed between the two sessions.

## Results and conclusion

We found, again, that the two digits could adapt simultaneously in opposite directions (*Appendix 2—figure 1A*). Thus, we were able to replicate the results of *Schot et al. (2014)* with different movements and a different perturbation (only feedback about the hand was displaced; when wearing prisms all visual information is displaced). Most importantly, the after-effect of the adaptation was just as clear in the trials with the digit-eye pairing that was *never* used during adaptation (open bars *Appendix 2—figure 1B*) as in the trials with the same digit-eye pairing as during adaptation (solid bars in *Appendix 2—figure 1B*). The lateral errors appear to be slightly lower than zero before the adaptation (leftmost part of *Appendix 2—figure 1A*). This is most likely a transfer of the after-effect of the previous session because the break between the two sessions was only a few minutes. The mean lateral error was indeed negligible for the first session that participants performed (95% confidence interval: −0.14 to 0.26 cm) but was −0.37 cm for the second session (95% confidence interval: −0.58 to −0.16 cm). As the order of the two kinds of sessions was randomized across participants, this transfer of after-effect did not influence the results.

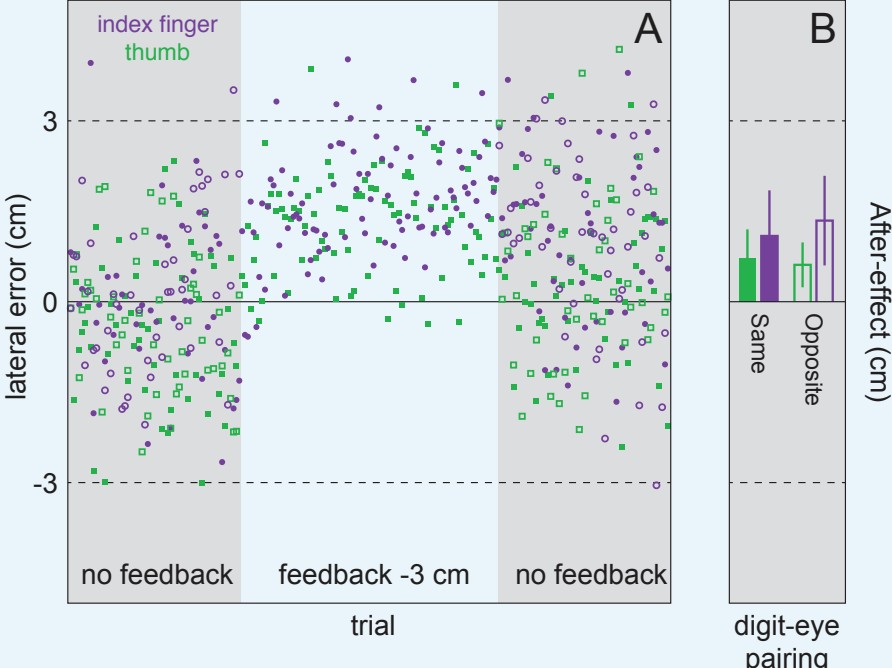

**Appendix 2—figure 1.** Data of the control experiment on the effect of digit-eye pairing. (**A**) The participants made pointing movements with either the unseen index finger (purple) or thumb (green) of their dominant hand (the right hand for 7 of the 8 participants). During adaptation (central part with a white background), participants received feedback about the endpoint of their movement. The feedback was shifted 3 cm to the left or the right. Before and after the adaptation trials they received no feedback (grey background). During adaptation, there was a fixed pairing between viewing eye and digit (filled symbols). Before and after the feedback trials, half the trials had the opposite pairing between viewing eye and digit (open symbols). Negative values on the y-axis represent the direction in which the feedback was shifted. So although the perturbation was always in opposite directions for the two digits (and sessions), the expected response is always in the positive direction. Each symbol indicates the median lateral error of all cases in which that digit was used in combination with that eye on that trial. (**B**) The after-effect of the adaptation: the mean

difference between the individual differences between the median lateral errors before and after feedback, separately for the thumb (green) and index finger (purple), and the same digit-eye pairing as was used during adaptation (filled bars) and the opposite pairing (open bars).
DOI: https://doi.org/10.7554/eLife.21440.011

We conclude that the simultaneous adaptation of digits of the same hand in opposite directions is indeed linked to the digit, rather than to the eye or a specific combination of digit and eye.

