## [Decision Letter]

Thank you for submitting your article "Unusual prism adaptation reveals how grasping is controlled" for consideration by *eLife*. Your article has been favorably evaluated by David Van Essen (Senior Editor) and three reviewers, one of whom, Richard Ivry (Reviewer #1), is a member of our Board of Reviewing Editors.

The reviewers have discussed the reviews with one another and the Reviewing Editor has drafted this decision to help you prepare a revised submission.

Summary:

We all find the paper intriguing. You have done a nice job of laying out three models and, as summarized in your table, describe how they make differential predictions concerning transfer from single digit training to a grasping task. It is clear that the results are inconsistent with the grasp and thumb control models. What is not clear to us is that the results provide a compelling demonstration that there is independent control of the digits, and that wrist position is simply an emergent property of this independent (yet coupled) form of control.

Essential revisions:

The first issue is that, as constructed, the transfer test does not distinguish your hypothesis (wrist control is emergent property) and a hypothesis in which the wrist position is also controlled (along with the fingers). This is because by using opposing directions of adaptation, the sum of these two would cancel out any generalization to the wrist. Thus, even if the wrist position was controlled during the transfer test, one would not expect to observe any displacement, the result you've obtained. It is not clear how you could distinguish between the emergent property account of the results and the cancellation account.

Second, because of your method, you end up with a confound in which adaptation is done under monocular conditions, separately for each eye, with transfer done under binocular conditions. I understand that this is because you have opted to reach for real objects and wanted to have a way to train each finger individually. However, the confound does mean that you could have created a situation in which the independence comes about because there is a unique mapping for each eye-effector combination. A priori, it is not clear that you need to use a monocular condition for the training. That is, you could have done this in binocular mode, using your current cueing method to instruct the participants to do the finger or thumb pointing movement (and applied the appropriate translation for each effector). If you go this route, you would still expect to see adaptation of each effector in the pointing task and, no wrist displacement during transfer. This would eliminate the confound.

The above is one experiment that would provide a novel test of your hypothesis without the monocular/binocular confound. However, we would still have the problem that the absence of wrist displacement may be due to "emergence" or cancellation, and I would like to see this addressed. Here is one idea to consider. Suppose you did training with just a single effector (let's do this binocular), and then did the transfer. You would, of course, predict that during transfer, the grip aperture would only be distorted for the trained effector. But what would happen to the wrist? I believe you would expect to see that the wrist position is displaced in the same direction as the change in the adapted effector (e.g., if right thumb is adapted outward, then wrist would also be pulled in that direction since it "tags along"). However, if there is control of the wrist, then one would expect to observe no change in wrist position, but just a distortion of the grip aperture. Should you run an experiment along these lines, you would want to compare transfer conditions in which they do single finger pointing (same as in adaptation phase) and grasping. This would allow you to directly compare the magnitude of the shift: Your model predicts it will be the same.

Along a similar line of thinking, it is interesting to ask what would happen if you adapted one finger and then did a transfer task where, instead of grasping the object, the participants were asked to point to the object with the fist. By your independent control hypothesis, one would assume no transfer under this condition (I assume the fist/hand can have its own independent controller.) I'm not certain that this result would be obtained. If you ran such an experiment, you would want to adapt/transfer in both directions (finger to fist and fist to finger), given the literature about generalization asymmetries between proximal and distal effectors.

I note that, with your unusual setup, it becomes interesting to know if there is transfer between eyes with your setup. That is, suppose you train the thumb with the left eye and then test with the right eye. My take is that you believe there would be good transfer here. However, if adaptation is somehow based on an effector-eye pairing, then there would be no transfer. This isn't directly relevant to your question. However, if one did not find transfer under these conditions, it would provide a different interpretation of your results.

I've thrown out a few issues here that need to be addressed in a revision. I do want to emphasize that it is essential to provide stronger tests of your independent control hypothesis. The results provide clear evidence that you can separately adapt two effectors. But this is really a confirmation of your earlier work and would not meet the *eLife* criteria for publication. The issue of how this impacts grasping is the key extension in the current paper and we do not feel the current evidence makes a clear case that this is an emergent property of the independent control of "two goal-directed single-digit movements."

[Editors' note: further revisions were requested prior to acceptance, as described below.]

Thank you for resubmitting your work entitled "Unusual prism adaptation reveals how grasping is controlled" for further consideration at *eLife*. Your revised article has been favorably evaluated by David Van Essen (Senior Editor), Rich Ivry (Reviewing Editor) and one of the original reviewers.

The manuscript has been improved but there are some remaining issues that need to be addressed before acceptance, as outlined below:

We are satisfied that the control experiment rules out the hypothesis that your independent adaptation effects are due to adaptation being specific to particular digit-eye mappings. The generalization across digit-eye mappings seems clear.

We continue to be concerned that the current design doesn't provide the strongest test of the independence hypothesis given that there is no predicted shift in wrist position due to the use of opposite signed shifts for the finger and thumb. As you outline, your theory predicts there should be no shift here. However, you are only providing the test here of a predicted null result. The positive result would involve a condition in which both the thumb and finger were shifted in the same direction. Here you would predict a shift in wrist position, whereas a theory in which the wrist was controlled would predict no shift. As we follow your argument in the response letter, you see your theory as agnostic in terms of how adaptation should affect body parts other than the thumb and finger (with wrist position as an emergent property). While this may be a good argument for not doing something like the finger-to-fist pointing experiment we had included in our suggestions, the same direction condition seems to be the strong test of your two core ideas, namely that grasping is the result of independent control of two effectors and that wrist position is an emergent property of this independent control.

We support *eLife*'s guidelines to minimize the number of back-and-forths in the review process and have taken this into consideration in our recommendation here. However, we think the inclusion of the same direction condition would really make a more convincing paper, and don't believe it is placing an excessive burden on you to make this request.

---

## [Author Response]

Essential revisions:The first issue is that, as constructed, the transfer test does not distinguish your hypothesis (wrist control is emergent property) and a hypothesis in which the wrist position is also controlled (along with the fingers). This is because by using opposing directions of adaptation, the sum of these two would cancel out any generalization to the wrist. Thus, even if the wrist position was controlled during the transfer test, one would not expect to observe any displacement, the result you've obtained. It is not clear how you could distinguish between the emergent property account of the results and the cancellation account.

The reviewers introduce here a new (fourth) hypothesis that has never been proposed: a theory in which three aspects are controlled: index finger in space, thumb in space, and wrist in space. Indeed, our data do not allow us to test this hypothesis, as wrist position was not measured. However, our results can clearly distinguish between the three hypotheses, as is explained in the table in the manuscript. We test the finger-thumb hypothesis against the thumb-grip hypothesis based on the transport position (and reject thumb-grip), and test the transport-grip hypothesis against the thumb-grip hypothesis based on the grip aperture (and reject transport-grip). We added a paragraph to the Introduction to explicitly clarify why we do not measure the movements of the wrist.

Second, because of your method, you end up with a confound in which adaptation is done under monocular conditions, separately for each eye, with transfer done under binocular conditions. I understand that this is because you have opted to reach for real objects and wanted to have a way to train each finger individually. However, the confound does mean that you could have created a situation in which the independence comes about because there is a unique mapping for each eye-effector combination. A priori, it is not clear that you need to use a monocular condition for the training. That is, you could have done this in binocular mode, using your current cueing method to instruct the participants to do the finger or thumb pointing movement (and applied the appropriate translation for each effector). If you go this route, you would still expect to see adaptation of each effector in the pointing task and, no wrist displacement during transfer. This would eliminate the confound.

The monocular training was used to be able to switch between prism orientations without having to switch the prisms. Switching the prisms between trials would take a considerable amount of extra time. Although we considered the eye-effector association to be very unlikely, we now tested this in a new control experiment. Such a digit-eye pairing should also apply to the single-digit after-effects as measured in Schot et al. 2014, and should not depend on using actual prisms, but hold for any sensorimotor learning based on monocular terminal feedback. As performing the adaptation experiment with real objects is rather labour-intensive, we used a virtual reality adaptation paradigm. We replicated the results of Schot et al., 2014, but now determining the aftereffect using monocular viewing. Moreover, we showed that the pairing between digit and eye played no role: for both digits, the aftereffect of adaptation was equally strong when viewing with either the same digit-eye pairing as used in the adaptation phase, or the opposite pairing. We have added this experiment as supplementary material.

The above is one experiment that would provide a novel test of your hypothesis without the monocular/binocular confound. However, we would still have the problem that the absence of wrist displacement may be due to "emergence" or cancellation, and I would like to see this addressed. Here is one idea to consider. Suppose you did training with just a single effector (let's do this binocular), and then did the transfer. You would, of course, predict that during transfer, the grip aperture would only be distorted for the trained effector. But what would happen to the wrist? I believe you would expect to see that the wrist position is displaced in the same direction as the change in the adapted effector (e.g., if right thumb is adapted outward, then wrist would also be pulled in that direction since it "tags along"). However, if there is control of the wrist, then one would expect to observe no change in wrist position, but just a distortion of the grip aperture. Should you run an experiment along these lines, you would want to compare transfer conditions in which they do single finger pointing (same as in adaptation phase) and grasping. This would allow you to directly compare the magnitude of the shift: Your model predicts it will be the same.

Again, the reviewers discuss the wrist, but that is beyond the scope of our study, as mentioned above. Specifically, our own hypothesis makes no predictions about the wrist because the wrist position is an emergent property, so we cannot critically test our hypothesis on the basis of the position of the wrist. However, it might be that the reviewers use “wrist” as a proxy for “grip position”. They might think that the two hypotheses predict the same (lack of) effect on grip position in our study, but different predictions for that variable in the proposed experiment. This is not the case. According to the finger-thumb hypothesis, you get adaptation of one finger, which will result in 50% adaptation in such a grip position (average of finger and thumb). For the transport-grip-hypothesis, the adaptation would involve a transport adaptation in half of the trials, so also possibly resulting in a 50% effect on grip position.

Along a similar line of thinking, it is interesting to ask what would happen if you adapted one finger and then did a transfer task where, instead of grasping the object, the participants were asked to point to the object with the fist. By your independent control hypothesis, one would assume no transfer under this condition (I assume the fist/hand can have its own independent controller.) I'm not certain that this result would be obtained. If you ran such an experiment, you would want to adapt/transfer in both directions (finger to fist and fist to finger), given the literature about generalization asymmetries between proximal and distal effectors.

Also for this issue, we argue that the models that we test do not make predictions of how adaptation transfers to body parts that are not directly involved.

I note that, with your unusual setup, it becomes interesting to know if there is transfer between eyes with your setup. That is, suppose you train the thumb with the left eye and then test with the right eye. My take is that you believe there would be good transfer here. However, if adaptation is somehow based on an effector-eye pairing, then there would be no transfer. This isn't directly relevant to your question. However, if one did not find transfer under these conditions, it would provide a different interpretation of your results.

We tested this in a separate experiment. As mentioned in our response to the reviewers’ second main issue, there is full transfer between the two eyes. This experiment is now reported.

I've thrown out a few issues here that need to be addressed in a revision. I do want to emphasize that it is essential to provide stronger tests of your independent control hypothesis. The results provide clear evidence that you can separately adapt two effectors. But this is really a confirmation of your earlier work and would not meet the eLife criteria for publication. The issue of how this impacts grasping is the key extension in the current paper and we do not feel the current evidence makes a clear case that this is an emergent property of the independent control of "two goal-directed single-digit movements."

The design of the experiment in the paper with real blocks and prisms was quite labour-intensive, which was necessary to study the effect of adapting pointing on grasping. In order to test the digit-eye association hypothesis, real objects are not necessary so we used a slightly different paradigm for this. As this control experiment is not directly related to the question, we present a detailed description in a supplementary file, and only mention the conclusion in the Discussion of the paper.

[Editors' note: further revisions were requested prior to acceptance, as described below.]

The manuscript has been improved but there are some remaining issues that need to be addressed before acceptance, as outlined below:We are satisfied that the control experiment rules out the hypothesis that your independent adaptation effects are due to adaptation being specific to particular digit-eye mappings. The generalization across digit-eye mappings seems clear.

We are glad that you like the control experiment.

We continue to be concerned that the current design doesn't provide the strongest test of the independence hypothesis given that there is no predicted shift in wrist position due to the use of opposite signed shifts for the finger and thumb. As you outline, your theory predicts there should be no shift here. However, you are only providing the test here of a predicted null result. The positive result would involve a condition in which both the thumb and finger were shifted in the same direction. Here you would predict a shift in wrist position, whereas a theory in which the wrist was controlled would predict no shift. As we follow your argument in the response letter, you see your theory as agnostic in terms of how adaptation should affect body parts other than the thumb and finger (with wrist position as an emergent property). While this may be a good argument for not doing something like the finger-to-fist pointing experiment we had included in our suggestions, the same direction condition seems to be the strong test of your two core ideas, namely that grasping is the result of independent control of two effectors and that wrist position is an emergent property of this independent control.

Unfortunately, we cannot follow the logic of this request. The reviewers suggest that for a condition in which both the thumb and finger were shifted in the same direction our theory “predicts a shift in wrist position, whereas a theory in which the wrist was controlled would predict no shift”. We would first of all like to point out that the reviewers are deriving predictions at the level of joints (the wrist) from our theory, whereas our theory explicitly does not address the joint level. We agree that it is not unreasonable to assume that the wrist will follow the digits to some extent, but the details of how the wrist will move are not part of our theory. Importantly, even if we make this assumption, the reviewers’ predictions seem to us to be incorrect, making the proposed experiment a bit meaningless.

We agree that it is likely that there will be a transfer of the common shift of both digits to the wrist, because if both digits are shifted in the same direction the wrist will probably follow. If we replace “wrist” by “transport”, which we define as the average of the two digits, we can explicitly consider our theory to make this prediction. Thus, according to our view, we predict transfer of adaptation of pointing that is common for both digits to the transport component of grasping. However, we do not see how a theory in which the wrist was controlled would predict no shift if both pointing movements are shifted in the same direction. If the wrist is shifted in the same manner to transport both the finger and the thumb when pointing, why would it not be shifted in the same way to transport the grip? As we understand it, the proponents of the control of “Grip aperture and wrist position” equate the transport of the wrist to reaching, so they would predict a transfer of reaching with a single digit to the transport component of grasping. To make sure that we have not just misunderstood the thumb-and-grasp control, we asked a proponent (Frank Zaal) what he would predict. He answered us that he would indeed predict the same outcome as we do. It therefore seems to us to be a waste of our and the participants’ time to conduct such a control.

We support eLife's guidelines to minimize the number of back-and-forths in the review process and have taken this into consideration in our recommendation here. However, we think the inclusion of the same direction condition would really make a more convincing paper, and don't believe it is placing an excessive burden on you to make this request.

We do not see how adding this condition would make a more convincing paper.